# Q-Learning for Stochastic Control under General Information Structures and Non-Markovian Environments

**Ali Devran Kara**  *alikara@umich.edu*
*Department of Mathematics*
*University of Michigan, Ann Arbor, MI, USA*

**Serdar Yüksel**  *yuksel@queensu.ca*
*Department of Mathematics and Statistics*
*Queen's University, Kingston, ON, Canada* *

Reviewed on OpenReview: *https: // openreview. net/ forum? id= 1Yp6xpTV55*

## Abstract

As our primary contribution, we present a convergence theorem for stochastic iterations, and in particular, Q-learning iterates, under a general, possibly non-Markovian, stochastic environment. Our conditions for convergence involve an ergodicity and a positivity criterion. We provide a precise characterization on the limit of the iterates and conditions on the environment and initializations for convergence. As our second contribution, we discuss the implications and applications of this theorem to a variety of stochastic control problems with non-Markovian environments involving (i) quantized approximations of fully observed Markov Decision Processes (MDPs) with continuous spaces (where quantization breaks down the Markovian structure), (ii) quantized approximations of belief-MDP reduced partially observable MDPS (POMDPs) with weak Feller continuity and a mild version of filter stability (which requires the knowledge of the model by the controller), (iii) finite window approximations of POMDPs under a uniform controlled filter stability (which does not require the knowledge of the model), and (iv) for multi-agent models where convergence of learning dynamics to a new class of equilibria, subjective Q-learning equilibria, will be studied. In addition to the convergence theorem, some implications of the theorem above are new to the literature and others are interpreted as applications of the convergence theorem. Some open problems are noted.

## 1 Introduction

In some stochastic control problems, one does not know the true dynamics or the cost structure, and may wish to use past data to obtain an asymptotically optimal solution (that is, via *learning* from past data). In some problems, this may be used as a numerical method to obtain approximately optimal solutions.

Yet, in many problems including most of those in health, applied and social sciences, and financial mathematics, one may not even know whether the problem studied can be formulated as a fully observed Markov Decision Process (MDP), or a partially observable Markov Decision Process (POMDP) or a multi-agent system where other agents are present or not. There are many practical settings where one indeed works with data and does not know the possibly very complex structure under which the data is generated and tries to respond to the environment.

A common practical and reasonable response is to view the system as an MDP, with a perceived state and action (which may or may not define a genuine controlled Markov chain and therefore, the MDP assumption may not hold in actuality), and arrive at corresponding solutions via some learning algorithm.

---

*This research was partially supported by the Natural Sciences and Engineering Research Council of Canada (NSERC)

The question then becomes two-fold: (i) Does such an algorithm converge? (ii) If it does, what does the limit mean for each of the following models: MDPs, POMDPs, and multi-agent models?

We answer the two questions noted above in the paper:

The answer to the first question will follow from a general convergence result, stated in Theorem 2.1 below. The result will require an ergodicity condition and a positivity condition, which will be specified and will need to be ensured depending on the specifics of the problem in various forms and initialization conditions. While our approach builds on Kara & Yüksel (2023a), the generality considered in this paper requires us to precisely present conditions on ergodicity and positivity, which will be important in applications.

The second question will entail further regularity and assumptions depending on the particular (hidden) information structure considered, whose implications under several practical and common settings will be presented.

Some of these have not been reported elsewhere and some will build on prior work though with a unified lens. We will first study fully observed MDPs with continuous spaces, then POMDPs with continuous spaces, and finally decentralized stochastic (or multi-agent) control problems.

We first show that under weak continuity conditions and a technical ergodicity condition, Q-learning can be applied to fully observable MDPs for near optimal performance under discounted cost criteria.

For POMDPs, we show that under a uniform controlled filter stability, finite memory policies can be used to arrive at near optimality, and with only asymptotic filter stability quantization can be used to arrive at near optimality, under a mild unique ergodicity condition (which entails the mild asymptotic filter stability condition) and weak Feller continuity of the non-linear filter dynamics. We note that the quantized approximations, for both MDPs and belief-MDPs, raise mathematical questions on unique ergodicity and the initialization, which are also addressed in the paper.

For decentralized stochastic control problems and multi-agent systems, under a variety of information structures with strictly local information or partial global information, we show that Q-learning can be used to arrive at equilibria, even though this may be a subjective one (i.e., one which may depend on subjective modeling or probabilistic assumptions of each agent).

We thus study reinforcement learning in stochastic control under a variety of models and information structures. The general theme is that reinforcement learning can be applied to a large class of models under a variety of performance criteria, provided that certain regularity conditions apply for the associated kernels.

We note that these questions and related ones have been studied in the literature starting with Singh et al. (1994), and Szepesvàri & Smart (2004); Melo et al. (2008), and including the recent studies Chandak et al. (2024) and Dong et al. (2022), to be discussed further below.

**Contributions.**

- The main contribution of the paper is Theorem 2.1, where we prove a general convergence result for a class of, possibly non-Markovian, stochastic iterations applicable to a large class of scenarios.

- In Section 3, we provide several applications and implications of Theorem 2.1. In particular, we show that Theorem 2.1 can be used to explain the convergence behavior observed in Q learning of several non-Markovian environments.

  (i) We note that the proof of Theorem 2.1, when applied to the standard finite model MDP setup, also offers an alternative to the standard martingale approach used to prove the convergence of Q learning (Jaakkola et al. (1994); Tsitsiklis (1994); see also Szepesvári (2010); Bertsekas & Tsitsiklis (1996a); Meyn (2022) for a general review). In particular, we do not require a separate proof for the boundedness of the iterates to establish the convergence result.

  (ii) Different versions Q learning under space discretization, and Q learning for POMDPs with finite memory information variables, as well as for multi agent systems have been shown to converge under various ergodicity assumptions. We show that Theorem 2.1 collectively explains these convergence results and also allows for further generalizations and relaxations: For example,

      in the context of Theorem 3.2 and Corollary 3.3, Theorem 2.1 allows us to relax the positive Harris recurrence assumption in (Kara & Yüksel, 2023a, Assumption 3) to only unique ergodicity which is a much more general condition especially for applications where the state process is uncountable.

(iii) In Section 3.4, we show that the application of Q learning for POMDPs with quantized (probability measure valued) belief states will converge under suitable assumptions. Note that this result is different from the use of finite memory history variables (utilized in Theorem 3.2 and Corollary 3.3) and is a new result. The complementary convergence conditions, building on the verification of Theorem 2.1, are stated in the paper with a detailed comparison noted in Remark 3.5.

(iv) The convergence result and its broad applicability raises an open question on the existence of equilibria where multiple agents learn their best response policies through subjectively updating their local approximate Q functions (i.e., responding via reinforcement learning under their subjective MDP models).

## 1.1 Convergence Notions for Probability Measures

For the analysis of the technical results, we will use different convergence and distance notions for probability measures.

Two important notions of convergences for sequences of probability measures are weak convergence, and convergence under total variation. For some $N \in \mathbb{N}$, a sequence $\{\mu_n, n \in \mathbb{N}\}$ in $\mathcal{P}(\mathbb{X})$ is said to converge to $\mu \in \mathcal{P}(\mathbb{X})$ *weakly* if $\int_{\mathbb{X}} c(x)\mu_n(dx) \to \int_{\mathbb{X}} c(x)\mu(dx)$ for every continuous and bounded $c : \mathbb{X} \to \mathbb{R}$. One important property of weak convergence is that the space of probability measures on a complete, separable, and metric (Polish) space endowed with the topology of weak convergence is itself complete, separable, and metric (Parthasarathy, 1967). One such metric is the bounded Lipschitz metric (Villani, 2008, p.109), which is defined for $\mu, \nu \in \mathcal{P}(\mathbb{X})$ as

$$\rho_{BL}(\mu, \nu) := \sup_{\|f\|_{BL} \leq 1} |\int f d\mu - \int f d\nu| \tag{1}$$

where

$$\|f\|_{BL} := \|f\|_\infty + \sup_{x \neq y} \frac{|f(x) - f(y)|}{d(x, y)}, \text{ and } \|f\|_\infty = \sup_{x \in \mathbb{X}} |f(x)|.$$

We next introduce the first order Wasserstein metric. The *Wasserstein metric* of order 1 for two distributions $\mu, \nu \in \mathcal{P}(\mathbb{X})$ is defined as

$$W_1(\mu, \nu) = \inf_{\eta \in \mathcal{H}(\mu, \nu)} \int_{\mathbb{X} \times \mathbb{X}} \eta(dx, dy)|x - y|,$$

where $\mathcal{H}(\mu, \nu)$ denotes the set of probability measures on $\mathbb{X} \times \mathbb{X}$ with the first marginal $\mu$ and the second marginal $\nu$. Furthermore, using the dual representation of the first order Wasserstein metric, we equivalently have

$$W_1(\mu, \nu) = \sup_{Lip(f) \leq 1} \left| \int f(x)\mu(dx) - \int f(x)\nu(dx) \right|$$

where $Lip(f)$ is the minimal Lipschitz constant of $f$.

A sequence $\{\mu_n\}$ is said to converge in $W_1$ to $\mu \in \mathcal{P}(\mathbb{X})$ if $W_1(\mu_n, \mu) \to 0$. For compact $\mathbb{X}$, the Wasserstein distance of order 1 metrizes the weak topology on the set of probability measures on $\mathbb{X}$ (see (Villani, 2008, Theorem 6.9)). For non-compact $\mathbb{X}$ convergence in the $W_1$ metric implies weak convergence (in particular this metric bounds from above the bounded-Lipschitz metric (Villani, 2008, p.109), which metrizes the weak convergence).

For probability measures $\mu, \nu \in \mathcal{P}(\mathbb{X})$, the *total variation* metric is given by

$$\|\mu - \nu\|_{TV} = 2 \sup_{B \in \mathcal{B}(\mathbb{X})} |\mu(B) - \nu(B)| = \sup_{f : \|f\|_\infty \leq 1} \left| \int f(x)\mu(dx) - \int f(x)\nu(dx) \right|,$$

where the supremum is taken over all measurable real $f$ such that $\|f\|_\infty = \sup_{x \in \mathbb{X}} |f(x)| \leq 1$. A sequence $\mu_n$ is said to converge in total variation to $\mu \in \mathcal{P}(\mathbb{X})$ if $\|\mu_n - \mu\|_{TV} \to 0$. We refer the reader to (Saldi et al., 2020, Sections 3 and 5) for further discussions on spaces of probability measures.

## 2 A Q-Learning Convergence Theorem under Non-Markovian Environments

Q-learning under non-Markovian settings have been studied recently in a few publications, e.g. Dong et al. (2022); Chandak et al. (2024); Kara & Yüksel (2023a). Prior to such recent studies, we note that Singh et al. (1994) showed the convergence of Q-learning for POMDPs with measurements viewed as state variables under certain conditions involving unique ergodicity of the hidden state process under the exploration. However, what the limit means for the original POMDP was not studied.

Recently, Kara & Yüksel (2023a) showed the convergence of Q-learning with finite window measurements and showed near optimality of the resulting limit under filter stability conditions. In the following, we will adapt the proof method in Kara & Yüksel (2023a) to the setup where the environment is an ergodic process but also generalize the class of problems for which the limit of the iterates can be shown to imply near optimality (for POMDPs) or near-equilibrium (for stochastic games).

The convergence result will serve complement to two highly related recent studies: Dong et al. (2022) and Chandak et al. (2024), but with both different contexts and interpretations as well as mathematical analysis.

Dong et al. (2022) presents a general paradigm of reinforcement learning under complex environments where an agent responds with the environment. A regret framework is considered, where the regret comparison is with regard to policies from a possibly suboptimal collection of policies. The variables are assumed to be finite, even though an infinite past dependence is allowed. The distortion measure for approximation is a uniform error among all past histories which are compatible with the presumed state. A uniform convergence result for the convergence of time averages is implicitly imposed in the paper. Chandak et al. (2024) considers the convergence of Q-learning under non-Markovian environments where the cost function structure is aligned with the paradigm in Dong et al. (2022). The setup in Chandak et al. (2024) assumes that the realized cost is a measurable function of the assumed finite state, a finite-space valued observable realization and the finite action; furthermore a continuity and measurability condition for the infinite dimensional observable process history is imposed which may be restrictive given the infinite dimensional history process and subtleties involved for such conditioning operations, e.g. in the theory of non-linear filtering processes Chigansky & Handel (2010). Chandak et al. (2024) pursues an ODE method for the convergence analysis (which was pioneered in Borkar & Meyn (2000)).

Regarding the comment on the infinite past dependence, the approximations in both Dong et al. (2022) and Chandak et al. (2024) require a worst case error in a sample-path sense, which, for example, is too restrictive for POMDPs, as it has been studied in Kara & Yüksel (2023a) and Kara & Yüksel (2022). Notably the term $L_t$ defined below in (10) is more natural and relaxed under filter stability conditions.

In our setup, there is an underlying model, the true incurred costs admit exogenous uncertainty which impacts the realized costs, and the considered hidden or observable random variables may be uncountable space valued. We adopt the general and concise proof method presented in Kara & Yüksel (2023a) (and Kara et al. (2023)), tailored to ergodic non-Markovian dynamics, to allow for convergence to a limit. The generality of our setup requires us to precisely present conditions for convergence.

Let $\{C_t\}_t$ be $\mathbb{R}$-valued, $\{S_t\}_t$ be $\mathbb{S}$-valued and $\{U_t\}_t$ be $\mathbb{U}$-valued three stochastic processes. Consider the following iteration defined for each $(s, u) \in \mathbb{S} \times \mathbb{U}$ pair

$$Q_{t+1}(s, u) = (1 - \alpha_t(s, u)) Q_t(s, u) + \alpha_t(s, u) (C_t + \beta V_t(S_{t+1})) \tag{2}$$

where $V_t(s) = \min_{u \in \mathbb{U}} Q_t(s, u)$, and $\alpha_t(s, u)$ is a sequence of constants also called the learning rates. We assume that the process $U_t$ is selected so that the following conditions hold. An umbrella sufficient condition is the following:

**Assumption 2.1.** $\mathbb{S}, \mathbb{U}$ *are finite sets, and the joint process* $(S_{t+1}, S_t, U_t, C_t)_{t \geq 0}$ *is asymptotically ergodic in the sense that for the given initialization random variable* $S_0$, *for any measurable bounded function* $f$, *we*

*have that with probability one,*

$$\frac{1}{N}\sum_{t=0}^{N-1} f(S_{t+1}, S_t, U_t, C_t) \to \int f(s_1, s, u, c)\pi(ds_1, ds, du, dc)$$

*for some measure $\pi$ such that $\pi(\mathbb{S} \times s \times u \times \mathbb{R}) > 0$ for any $(s,u) \in \mathbb{S} \times \mathbb{U}$.*

**Remark 2.1.** *Although we assume that $\mathbb{S}, \mathbb{U}$ are finite, we will sometimes use the integral sign instead of the summation sign for notational convenience and consistency, where we simply use the counting measure for finite spaces.*

**Remark 2.2.** *The assumption that $\pi(\mathbb{S} \times s \times u \times \mathbb{R}) > 0$ for any $(s,u) \in \mathbb{S} \times \mathbb{U}$ is in the same spirit as the standard condition for reinforcement algorithms that every state-action pair is visited infinitely often during training. We note that it is possible to relax this condition as we will see in Assumption 3.4, if one is only interested in the convergence of the algorithm. In particular, we might consider a measure $\pi$ such that $\pi(\mathbb{S} \times s \times u \times \mathbb{R}) > 0$ for all $(s,u) \in B \subset \mathbb{S} \times \mathbb{U}$, for some subset $B$, where the set $B$ represents so called trained state-action pairs. For the learned policies to be optimal, however, one needs to make sure that the controlled process stays within the trained part of the system during the execution of an optimal policy.*

The above implies Assumption 2.2(ii)-(iii) below:

**Assumption 2.2.** *i. $\alpha_t(s,u) = 0$ unless $(S_t, U_t) = (s,u)$. Furthermore,*

$$\alpha_t(s,u) = \frac{1}{1 + \sum_{k=0}^{t} 1_{\{S_k=s, U_k=u\}}}$$

*and with probability 1, $\sum_t \alpha_t(s,u) = \infty$*

*ii. For $C_t$, we have, as $t \to \infty$,*

$$\frac{\sum_{k=0}^{t} C_k 1_{\{S_k=s, U_k=u\}}}{\sum_{k=0}^{t} 1_{\{S_k=s, U_k=u\}}} \to C^*(s,u),$$

*almost surely for some function $C^*$.*

*iii. For the $S_t$ process, we have, for any function $f$, as $t \to \infty$,*

$$\frac{\sum_{k=0}^{t} f(S_{k+1}) 1_{\{S_k=s, U_k=u\}}}{\sum_{k=0}^{t} 1_{\{S_k=s, U_k=u\}}} \to \int f(s_1) P^*(ds_1|s,u)$$

*almost surely for some $P^*$.*

Note that a stationarity assumption is not required. Under Assumption 2.1, we have that with $f(S_{t+1}, S_t, U_t, C_t) = C_t 1_{\{S_t=s, U_t=u\}}$, as $N \to \infty$,

$$\frac{1}{N}\sum_{t=0}^{N-1} C_t 1_{\{S_t=s, U_t=u\}} \to \int_{C \in \mathbb{R}} C\pi(S=s, U=u, dC).$$

We also have that with $f(S_{t+1}, S_1, U_t, C_t) = 1_{\{S_t=s, U_t=u\}}$, as $N \to \infty$,

$$\frac{1}{N}\sum_{t=0}^{N-1} 1_{\{S_t=s, U_t=u\}} \to \pi(S=s, U=u)$$

almost surely. Hence, we can write

$$\frac{\frac{1}{t+1}\sum_{k=0}^{t} C_k 1_{\{S_k=s, U_k=u\}}}{\frac{1}{t+1}\sum_{k=0}^{t} 1_{\{S_k=s, U_k=u\}}} \to \int C\pi(dC|S=s, U=u) =: C^*(s,u)$$

which implies Assumption 2.2 (ii). Similarly, one can also establish Assumption 2.2 (iii) under Assumption 2.1.

As before, let $\mathbb{S}, \mathbb{U}$ be finite sets. Consider the following equation

$$Q^*(s, u) = C^*(s, u) + \beta \sum_{s_1 \in \mathbb{S}} V^*(s_1) P^*(s_1|s, u) \tag{3}$$

for some functions $Q^*$, $C^*$, to be defined explicitly, and for some regular conditional probability distribution $P^*(\cdot|s, u)$, where $V^*(u) := \min_u Q^*(s, u)$.

**Theorem 2.1.** *Under Assumption 2.2, $Q_t(s, u) \to Q^*(s, u)$ almost surely for each $(s, u) \in \mathbb{S} \times \mathbb{U}$ pair where $Q^*$ satisfies (3), for any initialization of $Q_0$.*

*Proof.* We adapt the proof method presented in (Kara & Yüksel, 2023a, Theorem 4.1), where instead of positive Harris recurrence, we build on ergodicity. We first prove that the process $Q_t$, determined by the algorithm in (2), converges almost surely to $Q^*$. We define

$$\begin{aligned}
\Delta_t(s, u) &:= Q_t(s, u) - Q^*(s, u) \\
F_t(s, u) &:= C_t + \beta V_t(S_{t+1}) - Q^*(s, u) \\
\hat{F}_t(s, u) &:= C^*(s, u) + \beta \sum_{s_1} V_t(s_1) P^*(s_1|s, u) - Q^*(s, u),
\end{aligned}$$

where $V_t(s) = \min_u Q_t(s, u)$.

Then, we can write the following iteration

$$\Delta_{t+1}(s, u) = (1 - \alpha_t(s, u)) \Delta_t(s, u) + \alpha_t(s, u) F_t(s, u).$$

Now, we write $\Delta_t = \delta_t + w_t$ such that

$$\begin{aligned}
\delta_{t+1}(s, u) &= (1 - \alpha_t(s, u)) \delta_t(s, u) + \alpha_t(s, u) \hat{F}_t(s, u) \\
w_{t+1}(s, u) &= (1 - \alpha_t(s, u)) w_t(s, u) + \alpha_t(s, u) r_t(s, u)
\end{aligned}$$

where $r_t := F_t - \hat{F}_t = \beta V_t(S_{t+1}) - \beta \sum_{s_1} V_t(s_1) P^*(s_1|s, u) + C_t - C^*(s, u)$. Next, we define

$$r_t^*(s, u) = \beta V^*(S_{t+1}) - \beta \sum_{s_1} V^*(s_1) P^*(s_1|s, u) + C_t - C^*(s, u)$$

We further separate $w_t = u_t + v_t$ such that

$$\begin{aligned}
u_{t+1}(s, u) &= (1 - \alpha_t(s, u)) u_t(s, u) + \alpha_t(s, u) e_t(s, u) \\
v_{t+1}(s, u) &= (1 - \alpha_t(s, u)) v_t(s, u) + \alpha_t(s, u) r_t^*(s, u)
\end{aligned}$$

where $e_t = r_t - r_t^*$.

We now show that $v_t(s, u) \to 0$ almost surely for all $(s, u)$. We have

$$v_{t+1}(s, u) = (1 - \alpha_t(s, u)) v_t(s, u) + \alpha_t(s, u) r_t^*(s, u).$$

When the learning rates are chosen such that $\alpha_t(s, u) = 0$ unless $(S_t, U_t) = (s, u)$, and,

$$\alpha_t(s, u) = \frac{1}{1 + \sum_{k=0}^{t} \mathbb{1}_{\{S_k=s, U_k=u\}}}$$

this term reduces to

$$v_{t+1}(s, u) = \frac{\sum_{k=0}^{t} r_k^*(s, u) \mathbb{1}_{\{S_k=s,, U_k=u\}} + v_0(s, u)}{1 + \sum_{k=0}^{t} \mathbb{1}_{\{S_k=s, U_k=u\}}}.$$

Recall that

$$r_k^*(s, u) = \beta V^*(S_{k+1}) - \beta \sum_{s_1} V^*(s_1) P^*(s_1|s, u) + C_k - C^*(s, u).$$

Hence, it is a direct implication of Assumption 2.2 that $v_t(s, u) \to 0$ almost surely for all $(s, u)$.

Now, we go back to the iterations:

$$\delta_{t+1}(s, u) = (1 - \alpha_t(s, u))\delta_t(s, u) + \alpha_t(s, u)\hat{F}_t(s, u)$$
$$u_{t+1}(s, u) = (1 - \alpha_t(s, u))u_t(s, u) + \alpha_t(s, u)e_t(s, u)$$
$$v_{t+1}(s, u) = (1 - \alpha_t(s, u))v_t(s, u) + \alpha_t(s, u)r_t^*(s, u).$$

Note that, we want to show $\Delta_t = \delta_t + u_t + v_t \to 0$ almost surely and we have that $v_t(s, u) \to 0$ almost surely for all $(s, u)$. The following analysis holds for any path that belongs to the probability one event in which $v_t(s, u) \to 0$. For any such path and for any given $\epsilon > 0$, we can find an $N < \infty$ such that $\|v_t\|_\infty < \epsilon$ for all $t > N$ as $(s, u)$ takes values from a finite set.

We now consider the term $\delta_t + u_t$ for $t > N$:

$$(\delta_{t+1} + u_{t+1})(s, u) = (1 - \alpha_t(s, u))(\delta_t + u_t)(s, u) + \alpha_t(s, u)(\hat{F}_t + e_t)(s, u). \tag{4}$$

Observe that for $t > N$,

$$(\hat{F}_t + e_t)(s, u) = (F_t - r_t^*)(s, u) = \beta V_t(S_{t+1}) - \beta V^*(S_{t+1}) \leq \beta \max_{s, u} |Q_t(s, u) - Q^*(s, u)| = \beta \|\Delta_t\|_\infty$$

$$\leq \beta \|\delta_t + u_t\|_\infty + \beta \epsilon$$

where the last step follows from the fact that $v_t \to 0$ almost surely. By choosing $C < \infty$ such that $\hat{\beta} := \beta(C+1)/C < 1$, for $\|\delta_t + u_t\|_\infty > C\epsilon$, we can write that

$$\beta \|\delta_t + u_t + \epsilon\|_\infty \leq \hat{\beta} \|\delta_t + u_t\|_\infty.$$

Now we rewrite (4)

$$(\delta_{t+1} + u_{t+1})(s, u) = (1 - \alpha_t(s, u))(\delta_t + u_t)(s, u) + \alpha_t(s, u)(\hat{F}_t + e_t)(s, u)$$
$$\leq (1 - \alpha_t(s, u))(\delta_t + u_t)(s, u) + \alpha_t(s, u)\hat{\beta} \|\delta_t + u_t\|_\infty \tag{5}$$
$$< \|\delta_t + u_t\|_\infty$$

Hence $\max_{s, u}((\delta_{t+1} + u_{t+1})(s, u))$ monotonically decreases for $\|\delta_t + u_t\|_\infty > C\epsilon$ and hence there are two possibilities: it either gets below $C\epsilon$ or it never gets below $C\epsilon$ in which case by the monotone non-decreasing property it will converge to some number, say $M_1$ with $M_1 \geq C\epsilon$.

First, we show that once the process hits below $C\epsilon$, it always stays there. Suppose $\|\delta_t + u_t\|_\infty < C\epsilon$,

$$(\delta_{t+1} + u_{t+1})(s, u) \leq (1 - \alpha_t(s, u))(\delta_t + u_t)(s, u) + \alpha_t(s, u)\beta (\|\delta_t + u_t\|_\infty + \epsilon)$$
$$\leq (1 - \alpha_t(s, u))C\epsilon + \alpha_t(s, u)\beta(C\epsilon + \epsilon)$$
$$= (1 - \alpha_t(s, u))C\epsilon + \alpha_t(s, u)\beta(C+1)\epsilon$$
$$\leq (1 - \alpha_t(s, u))C\epsilon + \alpha_t(s, u)C\epsilon, \quad (\beta(C+1) \leq C)$$
$$= C\epsilon.$$

To show that $M_1 \geq C\epsilon$ is not possible, we start by (5), we have that for all $(s, u)$

$$|(\delta_{k+1} + u_{k+1})(s, u)| \leq (1 - \alpha_k(s, u)) |(\delta_k + u_k)(s, u)| + \alpha_k(s, u)\hat{\beta} \|\delta_k + u_k\|_\infty$$

Assume $\|\delta_k + u_k\|_\infty$ is bounded by some $K_0 < \infty$ for all $k$, which we can always do since it is a decreasing sequence. One can then iteratively show that these iterations are bounded from above by the sequence solving the following dynamics

$$|\zeta_{k+1}(s, u)| = (1 - \alpha_k(s, u))\,|\zeta_k(s, u)| + \alpha_k(s, u)\hat{\beta}K_0.$$

Thus, $\zeta_k(s, u)$ will converge to the value $\hat{\beta}K_0$ for any initial point and starting from any time instance $k$. This follows since under the assumed learning rates, the iterates will converge to the averages of the constant $\hat{\beta}K_0$. Hence, the sequence $\|\delta_k + u_k\|_\infty$ will eventually become smaller than $\hat{\beta}K_0 + \kappa_0$ for any arbitrarily small $\kappa_0 > 0$. Similarly, once the sequence is bounded by some $K_1 := \hat{\beta}K_0 + \kappa_1$ (where $\kappa_1 > 0$ is arbitrarily small), they will eventually get smaller than $\hat{\beta}K_1 + \kappa$ for any $\kappa > 0$. Repeating the same argument, it follows that $\|\delta_k + u_k\|_\infty$ will hit below $C\epsilon$ eventually, in finite time.

This shows that the condition $\|\delta_t + u_t\|_\infty > C\epsilon$ cannot be sustained indefinitely for some fixed $C$ (independent of $\epsilon$). Hence, $(\delta_t + u_t)$ process converges to some value below $C\epsilon$ for any path that belongs to the probability one set. Then, we can write $\|\delta_t + u_t\|_\infty < C\epsilon$ for large enough $t$. Since $\epsilon > 0$ is arbitrary, taking $\epsilon \to 0$, we can conclude that $\Delta_t = \delta_t + u_t + v_t \to 0$ almost surely.

Therefore, the process $Q_t$, determined by the algorithm in (2), converges almost surely to $Q^*$. $\qquad\square$

## 2.1 An Example: Machine Replacement with non i.i.d. Noise

In this section we study the implications of the previous result on a machine replacement problem where the state process is not controlled Markov.

In this model, we have $\mathbb{X}, \mathbb{U}, \mathbb{W} = \{0, 1\}$ with

$$x_t = \begin{cases} 1 & \text{machine is working at time t} \\ 0 & \text{machine is not working at time t .} \end{cases} \qquad u_t = \begin{cases} 1 & \text{machine is being repaired at time t} \\ 0 & \text{machine is not being repaired at time t .} \end{cases}$$

We assume that the noise variable $w_t$ is not i.i.d. but is a Markov process with transition kernel

$$Pr(w_{t+1} = 0 | w_t = 0) = 0.9, \quad Pr(w_{t+1} = 0 | w_t = 1) = 0.4$$

Give the noise, we have the dynamics $x_{k+1} = f(x_k, u_k, w_k)$ for the controlled state process

$$
\begin{array}{ll}
x_1 = 0 \text{ if } x = 0, u = 0, w = 0, & x_1 = 0 \text{ if } x = 0, u = 0, w = 1 \\
x_1 = 1 \text{ if } x = 0, u = 1, w = 0, & x_1 = 0 \text{ if } x = 0, u = 1, w = 1 \\
x_1 = 1 \text{ if } x = 1, u = 0, w = 0, & x_1 = 0 \text{ if } x = 1, u = 0, w = 1 \\
x_1 = 1 \text{ if } x = 1, u = 1, w = 0, & x_1 = 0 \text{ if } x = 1, u = 1, w = 1
\end{array}
$$

In words, if the noise $w = 1$ then the machine breaks down at the next time step independent of the repair or the state of the machine. If the noise is not present, i.e. $w = 0$ then the machine is fixed if we decide to repair it, but stays broken if it was broken at the last step and we did not repair it.

The one stage cost function is given by

$$c(x, u) = \begin{cases} R + E & x = 0, u = 1 \\ E & x = 0, u = 0 \\ 0 & x = 1, u = 0 \\ R & x = 1, u = 1 \end{cases}$$

where $R$ is the cost of repair and $E$ is the cost incurred by a broken machine.

We study the example with discount factor $\beta = 0.7$, and $R = 1, E = 1.5$. For the exploration policy, we use a random policy such that $Pr(u_t = 0) = \frac{1}{2}$ and $Pr(u_t = 1) = \frac{1}{2}$ for all $t$.

Note that the state process $x_t$ is no longer a controlled Markov chain. However, we can check that Assumption 2.2 holds and that we can apply Theorem 2.1 to show that Q learning algorithm will converge. In particular, one can show that the joint process $(x_t, w_t)$ forms a Markov chain under the exploration policy, and admits a stationary distribution, say $\pi$ such that

$$\pi(x=0, w=0) = 0.145, \quad \pi(x=0, w=1) = 0.127, \quad \pi(x=1, w=0) = 0.654, \quad \pi(x=1, w=1) = 0.0728.$$

Hence, the Q learning algorithm constructed using $s_t = x_t$ will converge to the Q values of an MDP with the following transition probabilities:

$$P^*(s_1|s, u) = \frac{\sum_w \mathbb{1}_{\{s_1 = f(s,u,w)\}} \pi(s, u, w)}{\sum_w \pi(s, u, w)} \tag{6}$$

where $\pi(s, u, w) = \frac{1}{2}\pi(s, w)$ for the exploration policy we use.

In Figure 1, on the left we plot the learned value functions for the non-Markov state process when we take $s_t = x_t$. The plot on the right represents the learned value function when we simulate the environment as an MDP with the transition kernel $P^*$ given in (6). One can see that they converge to the same values as expected from the theoretical arguments.

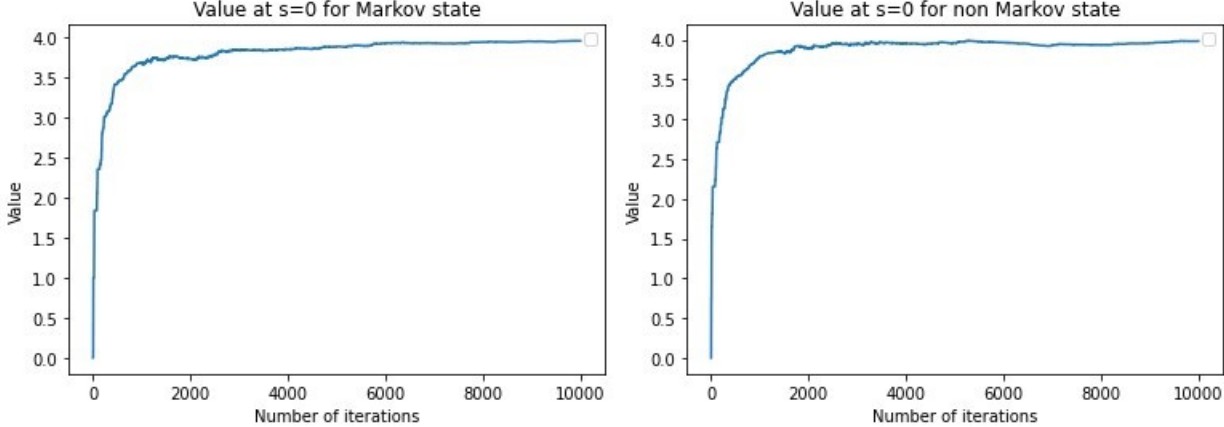

Figure 1: Q value convergence for non Markov and Markov state.

A further note is that since the state process is not Markov, the learned policies are not optimal. In particular, the Q leaning algorithm with $s_t = x_t$ learns the policy

$$\gamma_1(1) = 0, \quad \gamma_1(0) = 1.$$

Via simulation, the value of this policy can be found to be around 2.41 when the initial state, and the initial noise are uniformly distributed.

However, one can also construct the Q learning with $s_t = (x_t, x_{t-1})$ (which is still not a Markovian state, however). The learned policy in this case is

$$\gamma_2(s) = \begin{cases} 1 & \text{if } s = (0,0) \\ 0 & \text{otherwise} \end{cases}$$

The value of this policy can be simulated to be around 2.35, again when the initial state, and the initial noise are uniformly distributed. Thus, it performs better than the policy for the state variable $s_t = x_t$.

In the following, we will discuss a number of applications, together with conditions under which the limit of the iterates are near-optimal.

# 3 Implications and Applications under Various Information Structures

In this section, we study the implications and applications of the convergence result in Theorem 2.1; some of the applications and refinements are new and some are from recent results viewed in a unified lens.

We first start with a brief review involving Markov Decision Processes. Consider the model $x_{t+1} = f(x_t, u_t, w_t)$, where $x_t$ is an $\mathbb{X}$-valued state variable, $u_t$ a $\mathbb{U}$-valued control action variable, $w_t$ a $\mathbb{W}$-valued i.i.d noise process, and $f$ a function, where $\mathbb{X}, \mathbb{U}, \mathbb{W}$ are appropriate spaces, defined on some probability space $(\Omega, \mathcal{F}, P)$. By, e.g. (Gihman & Skorohod, 2012, Lemma 1.2), the model above contains processes satisfying the following for all Borel $B$ and $t \geq 0$

$$P(x_{t+1} \in B | x_{[0,t]} = a_{[0,t]}, u_{[0,t]} = b_{[0,t]}) = P(x_{t+1} \in B | x_t = a_t, u_t = b_t) =: \mathcal{T}(B | a_t, b_t) \tag{7}$$

where $\mathcal{T}(\cdot | x, u)$ is a *stochastic kernel* from $\mathbb{X} \times \mathbb{U}$ to $\mathbb{X}$. Here, $x_{[0,t]} := \{x_0, x_1, \cdots, x_t\}$. A stochastic process which satisfies (7) is called a *controlled Markov chain*. Let the control actions $u_t$ be generated via a control policy $\gamma = \{\gamma_t, t \geq 0\}$ with $u_t = \gamma_t(I_t)$, where $I_t$ is the information available to the Decision Maker (DM) or controller at time $t$. If $I_t = \{x_0, \cdots, x_t; u_0, \cdots, u_{t-1}\}$, we have a *fully observed* system and an optimization problem is referred to as a *Markov Decision Process (MDP)*. As an optimization criterion, given a cost function $c : \mathbb{X} \times \mathbb{U} \to \mathbb{R}_+$, one may consider $J_\beta(x, \gamma) = E_x^\gamma[\sum_{t=0}^\infty \beta^t c(x_t, u_t)]$, for some $\beta \in (0, 1)$ and $x_0 = x$. This is called a *discounted infinite-horizon optimal control problem* Bertsekas (1976).

If the DM has only access to noisy measurements $y_t = g(x_t, v_t)$, with $v_t$ being another i.i.d. noise, and $I_t = \{y_0, \cdots, y_t; u_0, \cdots, u_{t-1}\}$, we then have a *Partially Observable Markov Decision Process (POMDP)*. We let $O(y_t \in \cdot | x_t = x)$ denote the transition kernel for the measurement variables. We will assume that $c$ is continuous and bounded, though the boundedness can be relaxed.

We assume in the following that $\mathbb{X}$ is a compact subset of a Polish space and that $\mathbb{Y}$ is finite. We assume that $\mathbb{U}$ is a compact set. However, without any loss, but building on (Saldi et al., 2018, Chapter 3), under weak Feller continuity conditions (i.e., $E[f(x_1)|x_0 = x, u_0 = u]$ is continuous in $(x, u) \in \mathbb{X} \times \mathbb{U}$ for every bounded continuous $f$), we can approximate $\mathbb{U}$ with a finite set with an arbitrarily small performance loss. Accordingly, we will assume that this set is finite.

The same applies when a POMDP is reduced to a belief-MDP and the belief-MDP is weak Feller: POMDPs can be reduced to a completely observable Markov process Yushkevich (1976); Rhenius (1974), whose states are the posterior state distributions or *beliefs*:

$$\pi_t(\cdot) := P\{X_t \in \cdot | Y_0, \ldots, Y_t, U_0, \ldots, U_{t-1}\} \in \mathcal{P}(\mathbb{X}),$$

where $\mathcal{P}(\mathbb{X})$ is the set of probability measures on $\mathbb{X}$. We call $\pi_t$ the filter process, whose transition probability can be constructed via a Bayesian update. With

$$F(\pi, u, y) := P\{X_{t+1} \in \cdot | \pi_t = \pi, u_t = u, y_{t+1} = y\},$$

and the stochastic kernel

$$K(\cdot | \pi, u) = P\{y_{t+1} \in \cdot | \pi_t = \pi, u_t = u\},$$

we can write a transition kernel, $\eta$, for the belief process:

$$\eta(\cdot | \pi, u) = \int_\mathbb{Y} 1_{\{F(\pi, u, y) \in \cdot\}} K(dy | \pi, u). \tag{8}$$

The equivalent cost function is

$$\tilde{c}(\pi_t, u_t) := \int_\mathbb{X} c(x, u_t) \pi_t(dx).$$

Thus, the filter process defines a *belief*-MDP with kernel $\eta$. The kernel $\eta$ is a weak Feller kernel if (a) If $\mathcal{T}$ is weakly continuous and the measurement kernel $O(y_t \in \cdot | x_t = x)$ is total variation continuous Feinberg et al. (2016) (see also Crisan & Doucet (2002)), or (b) if the kernel $\mathcal{T}$ is total variation continuous (with no assumptions on $O$) Kara et al. (2019).

Wasserstein regularity of $\eta$ is studied in Demirci et al. (2023). We will also consider multi-agent models, to be discussed further below.

Recall (3) which is the limit of the Q iterates if they converge:

$$Q^*(s, u) = C^*(s, u) + \beta \sum_{s_1 \in \mathbb{S}} V^*(s_1) P^*(s_1 | s, u).$$

We note that these Q values correspond to an MDP model with state space $\mathbb{S}$, the action space $\mathbb{U}$, the stage-wise cost function is $C^*(s, u)$ and the transition model is $P^*(s_1 | s, u)$. Hence, the policies constructed using these Q values are optimal for the corresponding MDP. In the following, we will present some bounds in terms of how 'close' the original control model, and the approximate MDP model the limit Q values correspond to are.

## 3.1 Finite MDPs

Consider a finite MDP where the state process $x_t$ takes values in some finite set $\mathbb{X}$, the control action process $u_t$ takes values in some finite set $\mathbb{U}$. The dynamics for the state process is governed by the following

$$x_{t+1} \sim P^*(\cdot | x_t, u_t)$$

and at each time $t$, the controller receives a stage-wise cost

$$c(x_t, u_t).$$

**Corollary 3.1** (Corollary to Theorem 2.1). *The iterations in (2) converges a.s. with $S_t = x_t$ and $C_t = c(x_t, u_t)$, if the learning rates $\alpha_t$ satisfy Assumption 2.2(i). Furthermore, the limit $Q^*$ is the optimal Q values of the system.*

**Remark 3.1.** *This, of course, is a standard result Watkins & Dayan (1992); Tsitsiklis (1994); Baker (1997); Szepesvári & Littman (1999); Szepesvári (2010); Bertsekas & Tsitsiklis (1996b). However, we emphasize that different from the standard martingale proof (e.g. Jaakkola et al. (1994); Tsitsiklis (1994)), we do not need to separately establish the boundedness of the iterates due to the convergence property. Accordingly, the above can also be seen as an alternative proof of the standard Q-learning algorithm, though we restrict the exploration policy (unlike the standard proof where such a restriction is not needed as long as each state action pair is visited infinitely often with no ergodicity condition, together with standard summability conditions). We also note that avoiding the boundedness of the iterates is essential to extend the result to non-Markovian environments.*

## 3.2 Quantized Q-Learning for Weakly Continuous MDPs with General Spaces

In this section, we assume that $\mathbb{X}$ is a compact subset of a Polish space and that $\mathbb{Y}$ and $\mathbb{U}$ are finite sets. We use $d(x, x')$ to metrize the space $\mathbb{X}$ for any $x, x' \in \mathbb{X}$.

Consider a controlled Markov chain $X_t$ whose dynamics are determined by

$$X_{t+1} \sim \mathcal{T}(\cdot | x_t, u_t).$$

Furthermore, let $C_t := c(X_t, U_t)$ take values from a bounded set. The controller observes the cost realizations and some noisy version of the hidden state variable. In particular, we assume that the controller observes the measurement process $Y_t$ as

$$Y_t = g(X_t, V_t) \tag{9}$$

for some measurable function $g$ and for some i.i.d. noise process $V_t$.

In the following, we let the measurement structure be so that it corresponds to a quantization of the state variable $X_t$: We discretize continuous MDPs, where the state space $\mathbb{X}$ is quantized such that for disjoint $\{B_i\}_{i=1}^M$ with $\cup_{i=1}^M B_i = \mathbb{X}$, we define a finite set $\mathbb{S} = \{y_1, \ldots, y_m\}$ and write

$$y_i = g(x), \text{ if } x \in B_i.$$

We take $S_k = g(X_k) = Y_k$. Therefore, the problem can be seen as a POMDP and thus an adaptation of Assumption 2.1 will guarantee the convergence of the iterations in (2). In particular, we present the following assumption that implies Assumption 2.1 in the context of quantized MDPs.

**Assumption 3.1.** *Under the exploration policy $\gamma$ and initialization, the controlled state and control action joint process $\{X_t, U_t\}$ is asymptotically ergodic in the sense that for any measurable bounded function $f$ we have that*

$$\lim_{N \to \infty} \frac{1}{N} \sum_{t=0}^{N-1} f(X_t, U_t) = \int f(x, u) \pi^\gamma(dx, du)$$

*for some $\pi^\gamma \in \mathcal{P}(\mathbb{X} \times \mathbb{U})$ such that $\pi^\gamma(B_i \times u) > 0$ for any quantization bin $B_i$ and for any $u \in \mathbb{U}$.*

We note that a sufficient condition for the ergodicity assumption, for every initialization of $X_0$, would be positive Harris recurrence under the exploration policy.

**Corollary 3.2** (Corollary to Theorem 2.1)**.** *The iterations given in (2) converges almost surely under Assumption 3.1 and Under Assumption 2.2 (i) with $S_k = g(X_k) = Y_k$ and $C_k = c(X_k, U_k)$.*

The limit Q values correspond to an approximate control model (see Kara et al. (2023)). For near optimality of the learned polices (Kara et al., 2023, Corollary 12) notes the following:

**Assumption 3.2.**     *(a)  $\mathbb{X}$ is compact.*

  *(b)  There exists a constant $\alpha_c > 0$ such that $|c(x, u) - c(x', u)| \leq \alpha_c d(x, x')$ for all $x, x' \in \mathbb{X}$ and for all $u \in \mathbb{U}$.*

  *(c)  There exists a constant $\alpha_T > 0$ such that $W_1(\mathcal{T}(\cdot|x, u), \mathcal{T}(\cdot|x', u)) \leq \alpha_T d(x, x')$ for all $x, x' \in \mathbb{X}$ and for all $u \in \mathbb{U}$.*

**Theorem 3.1.** *(Kara et al., 2023, Corollary 12)*

  *(a)  Let Assumption 3.2 hold . Then, for the policy constructed from the limit Q values, say $\hat{\gamma}$, we have*

$$\sup_{x_0 \in \mathbb{X}} \left| J_\beta(x_0, \hat{\gamma}) - J_\beta^*(x_0) \right| \leq \frac{2\alpha_c}{(1 - \beta)^2 (1 - \beta \alpha_T)} \bar{L}.$$

  *where*

$$\bar{L} := \max_{i=1,\dots,M} \sup_{x,x' \in B_i} d(x, x').$$

  *(b)  Saldi et al. (2017) For asymptotic convergence (without a rate of convergence) to optimality as the quantization rate goes to $\infty$, only weak Feller property of $\mathcal{T}$ is sufficient for the the algorithm to be near optimal.*

**Remark 3.2.** *Further error bounds under different set of assumptions, such as for systems with non-compact state space $\mathbb{X}$ and non-uniform quantization and models with total variation continuous transition kernels can be found in Kara et al. (2023). Q-learning for average cost problems involving continuous space models has recently been studied in Kara & Yüksel (2023b).*

### 3.3   Finite Window Memory POMDP with Uniform Geometric Controlled Filter Stability

We now assume that $\mathbb{X}$ is a compact subset of a Polish space and that $\mathbb{Y}$ and $\mathbb{U}$ are finite sets.

Suppose that the controller keeps a finite window of the most recent $N$ observation and control action variables, and perceives this as the *state* variable, which is in general non-Markovian. That is we take

$$S_t = \{Y_{[t-N,t]}, U_{[t-N,t-1]}\},$$

and $C_t := c(X_t, U_t)$.

In this case, the pair $(S_t, X_t, U_t)$ forms a controlled Markov chain, even if $(S_t, U_t)$ does not. We state the ergodicity assumption formally next.

**Assumption 3.3.** *(i) Under the exploration policy $\gamma$ and initialization, and the controlled state and control action joint process $\{X_t, U_t\}$ is asymptotically ergodic in the sense that for any measurable bounded function $f$ we have that*

$$\lim_{N \to \infty} \frac{1}{N} \sum_{t=0}^{N-1} f(X_t, U_t) = \int f(x,u) \pi^\gamma(dx, du)$$

*for some $\pi^\gamma \in \mathcal{P}(\mathbb{X} \times \mathbb{U})$. Furthermore, we have that $P(Y_t = y|x) > 0$ for every $x \in \mathbb{X}$.*

*(ii) Assumption 2.1(i) holds with $S_t = \{Y_{[t-N,t]}, U_{[t-N,t-1]}\}$.*

We note that a sufficient condition for the ergodicity assumption, for every initialization of $X_0$, would be positive Harris recurrence under the exploration policy.

**Corollary 3.3.** *[Corollary to Theorem 2.1] Under Assumption 3.3 and Assumption 2.2(i), the iterations in (2) converges a.s. with $S_t = \{Y_{[t-N,t]}, U_{[t-N,t-1]}\}$ and $C_t := c(X_t, U_t)$.*

The question then is whether the limit Q values correspond to a meaningful control problem, and how 'close' this control problem is to the original POMDP. We denote by $J_\beta(\pi_N^-, \mathcal{T}, \gamma^N)$ the value of the partially observed control problem when the initial prior measure of the hidden state $X_N$ at time $N$ is given by $\pi_N^-$ and when we use finite window control policy. In particular, the costs are incurred after the $N$-measurements are collected. (Kara & Yüksel, 2023a, Theorem 4.1) shows that the limit Q values indeed correspond to an approximate control problem, and notes the following bound on the optimality gap for the finite window control policies:

**Theorem 3.2.** *(Kara & Yüksel, 2023a, Theorem 4.1) If we denote the policies constructed using the limit Q values by $\gamma^N$, and apply $\gamma^N$ in the original problem, we obtain the following error bound:*

$$E\left[J_\beta(\pi_N^-, \mathcal{T}, \gamma^N) - J_\beta^*(\pi_N^-, \mathcal{T})|I_0^N\right] \leq \frac{2\|c\|_\infty}{(1-\beta)} \sum_{t=0}^{\infty} \beta^t L_t$$

*where $I_0^N$ is the first $N$ observation and control variables, that is*

$$I_0^N = \{Y_0, \ldots, Y_N, U_0, \ldots, U_{N-1}\}$$

*and the expectation is taken with respect to different realizations of $I_0^N$ under the initial distribution of the hidden state $\pi_0$ and the initialization policy used in the first $N$ time steps. Furthermore,*

$$\pi_N^- = P(X_N \in \cdot|I_0^N)$$

*and*

$$L_t := \sup_{\hat{\gamma} \in \hat{\Gamma}} E_{\pi_0^-}^{\hat{\gamma}} \left[\|P^{\pi_t^-}(X_{t+N} \in \cdot|Y_{[t,t+N]}, U_{[t,t+N-1]}) - P^{\pi^*}(X_{t+N} \in \cdot|Y_{[t,t+N]}, U_{[t,t+N-1]})\|_{TV}\right] \tag{10}$$

*and $\pi^*$ is the invariant measure on $x_t$ under the exploration policy $\gamma$.*

**Remark 3.3.** *Theorem 2.1 allows us to relax the positive Harris recurrence assumption in (Kara & Yüksel, 2023a, Assumption 3) to only unique ergodicity which is a significantly more relaxed condition for applications where the state process is uncountable.*

**Remark 3.4.** *The term $L_t$ is related to the filter stability problem, and explicit bounds for this can be found in (Kara & Yüksel, 2023a, Section 5), notably building on McDonald & Yüksel (2020): Recall the following*

**Definition 3.1.** *(Dobrushin, 1956, Equation 1.16) For a kernel operator $K : S_1 \to \mathcal{P}(S_2)$ (that is a regular conditional probability from $S_1$ to $S_2$) for standard Borel spaces $S_1, S_2$, we define the Dobrushin coefficient as:*

$$\delta(K) = \inf \sum_{i=1}^{n} \min(K(x, A_i), K(y, A_i)) \tag{11}$$

*where the infimum is over all $x, y \in S_1$ and all partitions $\{A_i\}_{i=1}^n$ of $S_2$.*

*This definition holds for both continuous or finite/countable spaces $S_1$ and $S_2$ and $0 \leq \delta(K) \leq 1$ for any kernel operator. Let*

$$\tilde{\delta}(\mathcal{T}) := \inf_{u \in \mathbb{U}} \delta(\mathcal{T}(\cdot|\cdot, u)).$$

*We then have that (Kara & Yüksel, 2023a, Section 5)*

$$L_t \leq ((1 - \tilde{\delta}(\mathcal{T}))(2 - \delta(O)))^N.$$

### 3.4 Quantized Approximations for Weak Feller POMDPs with only Asymptotic Filter Stability

We again assume that $\mathbb{X}$ is a compact subset of a Polish space and that $\mathbb{Y}$ and $\mathbb{U}$ are finite sets.

As noted earlier, any POMDP can be reduced to a completely observable Markov process (Yushkevich (1976), Rhenius (1974)) (see (8)), whose states are the posterior state distributions or *belief*s of the observer; that is, the state at time $t$ is

$$\pi_t(\,\cdot\,) := P\{X_t \in \cdot \,|y_0, \ldots, y_t, u_0, \ldots, u_{t-1}\} \in \mathcal{P}(\mathbb{X}).$$

We call this conditional probability measure process the *filter* process.

Recall the kernel $\eta$ (8) for the filter process. Now, by combining the quantized Q-learning above in Section 3.2 and the weak Feller continuity results for the non-linear filter kernel (Feinberg et al. (2016) Kara et al. (2019)), we can conclude that the setup in Section 3.2 is applicable though with a significantly more tedious analysis involving ergodicity requirements. Additionally, one needs to quantize probability measures (that is, beliefs or filter realizations). Accordingly, we take $S_t = g(\pi_t)$ for some quantizer

$$g : \mathcal{P}(\mathbb{X}) \to \mathcal{P}(\mathbb{X})^M =: \{B_1, B_2, \cdots, B_{|\mathcal{P}(\mathbb{X})^M|}\},$$

with $|\mathcal{P}(\mathbb{X})^M| < \infty$, and $C_t := c(X_t, U_t)$.

We state the ergodicity condition formally:

**Assumption 3.4.** *Under the exploration policy $\gamma$ and initialization, the controlled belief state and control action joint process $\{\pi_t, U_t\}$ is asymptotically uniquely ergodic in the sense that for any measurable bounded function $f$ we have that*

$$\lim_{N \to \infty} \frac{1}{N} \sum_{t=0}^{N-1} f(\pi_t) = \int f(\pi) \eta^\gamma(d\pi)$$

*for some $\eta^\gamma \in \mathcal{P}(\mathcal{P}(\mathbb{X}) \times \mathbb{U})$.*

We refer to the set

$$\mathcal{P}_\eta := \{\pi : \pi \in B_i \subset \mathcal{P}(\mathbb{X}) : \eta^\gamma(B_i) > 0\},$$

as the trained set of states; since these sets will be visited infinitely often under the exploration policy.

Unique ergodicity of the dynamics follows from results in the literature, such as, (Masi & Stettner, 2005, Theorem 2) and (van Handel, 2009, Prop 2.1), which hold when the randomized control is memoryless under mild conditions on the process notably that the hidden variable is a uniquely ergodic Markov chain and the measurement structure satisfies filter stability in total variation in expectation (one can show that weak merging in expectation also suffices); we refer the reader to (McDonald & Yüksel, 2024, Figure 1) for mild conditions leading to filter stability in this sense, which is related to stochastic observability (McDonald & Yüksel, 2024, Definition II.1). Notably, a uniform and geometric controlled filter stability is not required even though this would be sufficient. Therefore, due to the weak Feller property of controlled non-linear filters, we can apply the Q-learning algorithm to also belief-based models to arrive at near optimal control policies. Nonetheless, since positive Harris recurrence cannot be assumed for the filter process, the initial state of the filter process may not be arbitrary: If the invariant measure under the exploration policy is the initial state (of the filter process), (van Handel, 2009, Prop 2.1) implies that the time averages will converge as imposed in Assumption 2.2. A sufficient condition for unique ergodicity then is the following.

**Assumption 3.5.** *Under the exploration policy $\gamma$ the hidden process $\{X_t\}$ is uniquely ergodic (with measure $\zeta$) and the measurement dynamics are so that the filter is stable in expectation under weak convergence.*

The initialization during the implementation of the algorithm affects both the trained sets, which are those visited infinitely often, and the computation of policies for the set of states reachable from the initialization: The condition that $\eta^\gamma(B_i) > 0$ requires an analysis tailored for each problem. For example, if the quantization is performed as in Kara & Yüksel (2022) by clustering bins based on a finite past window, then the condition is satisfied by requiring that $P(Y_t = y|x) > 0$ for every $x \in \mathbb{X}$. If the clustering is done, e.g. by quantization of the probability measures via first quantizing $\mathbb{X}$ and then quantizing the probability measures on the finite set (see (Saldi et al., 2020, Section 5)), then the initialization could be done according to the invariant probability measure corresponding to the hidden Markov source.

For some applications, the quantization does not have to be uniform as the entire probability space $\mathcal{P}(\mathbb{X})$ may not be visited; in this case it suffices to have the conditions be restricted to the subset reachable from the initial probability measure under any policy (or at least a set of policies which contains an optimal policy). Accordingly, if during the implementation for a given initialization one can ensure that all the visited states while applying an optimal policy remain inside $\mathcal{P}_\eta$, near optimality follows. A sufficient condition is that $\pi_0 \sim \kappa \ll \eta^\gamma$ or that $\pi_0$ is in the topological support of $\eta^\gamma$; see (Cregg et al., 2023, Corollary 2).

**Assumption 3.6.** *The controlled transition kernel for the belief process $\eta(\cdot|\pi, u)$ is Lipschitz continuous under the metric $W_1$ such that*

$$W_1\left(\eta(\cdot|z, u), \eta(\cdot|z', u)\right) \leq \alpha_T W_1(z, z')$$

*for all $u$, and $z, z' \in \mathcal{P}(\mathbb{X})$ for some $\alpha_T < \infty$.*

The following result from (Demirci et al., 2023, Theorem 2.3) provides a set of assumptions on the partially observed model to guarantee Assumption 3.6 when $\mathcal{P}(\mathbb{X})$ is equipped with the $W_1$ metric.

**Proposition 3.1.** *(Demirci et al., 2023, Theorem 2.3)*

1. *$(\mathbb{X}, d)$ is a compact metric space with diameter $D$ (where $D = \sup_{x,y \in \mathbb{X}} d(x, y)$).*

2. *There exists $\alpha \in R^+$ such that*

$$\|\mathcal{T}(\cdot \mid x, u) - \mathcal{T}(\cdot \mid x', u)\|_{TV} \leq \alpha d(x, x')$$

*for every $x, x' \in \mathbb{X}$, $u \in \mathbb{U}$.*

*Under the conditions above we have*

$$W_1\left(\eta(\cdot \mid z_0, u), \eta\left(\cdot \mid z_0', u\right)\right) \leq \left(\frac{\alpha D(3 - 2\delta(O))}{2}\right) W_1\left(z_0, z_0'\right).$$

*for all $z_0, z_0' \in \mathcal{Z}$, $u \in \mathbb{U}$.*

Thus, Assumption 3.6 holds with $\alpha_T = \frac{\alpha D(3 - 2\delta(O))}{2}$.

**Theorem 3.3.**     *(a) Suppose that under the exploration policy and initialization the controlled filter process satisfies Assumption 3.4 and 2.2(i) with $S_t = g(\pi_t)$, and $C_t = c(X_t, U_t)$. Then, the $Q$ iterates converge almost surely.*

    *(b) Let Assumption 3.6 hold such that $\alpha_T \beta < 1$ and assume that the cost function $c(x, u)$ is Lipschitz continuous in $x$ such that*

$$|c(x, u) - c(x', u)| \leq \alpha_c d(x, x').$$

    *For the policy constructed using the limit $Q$ values, say $\hat{\gamma}$ we have the following bound:*

$$J_\beta^*(\pi_0, \hat{\gamma}) - J_\beta^*(\pi_0) \leq \frac{2\alpha_c}{(1 - \beta)^2(1 - \beta\alpha_T)} \bar{L}.$$

where

$$\bar{L} := \sup_{\pi} W_1(\pi, g(\pi)) \tag{12}$$

*(c) The bound in (b) above also holds if (12) is relaxed to*

$$\bar{L} := \sup_{\pi \in \mathrm{supp}(\eta^\gamma), \pi \in B_i : \eta^\gamma(B_i) > 0} W_1\left(\pi, g(\pi)\right),$$

*provided that $\pi_0$ is in the support of $\eta^\gamma$ (such as $\pi_0$ being the invariant measure of $X_t$ under the exploration policy).*

*(d) For asymptotic convergence (without a rate of convergence) to optimality as the quantization rate goes to $\infty$ (i.e., $\bar{L} \to 0$), only weak Feller property of $\eta$ is sufficient for the the algorithm to be near optimal.*

Notably, suppose that for exploration $\pi_0 \sim \kappa \ll \phi$ or $\pi_0 = \zeta$ where $\zeta$ is the invariant measure for the hidden state process under exploration. Then, the $Q_t$ iterates converge almost surely.

The above approximation result, given the general convergence theorem, Theorem 2.1, follows from Kara et al. (2023)[Theorem 6,7] under the provided assumptions.

**Remark 3.5.** *We now present a comparison between the two approaches given in Sections 3.3 and 3.4 above: filter quantization vs. finite window based learning:*

*(a) (i) For the filter quantization, we only need unique ergodicity of the filter process under the exploration policy for which asymptotic filter stability in expectation in weak or total variation is sufficient.*

*(ii) The running cost can start immediately without waiting for a window of measurements.*

*(iii) On the other hand, the controller must run the filter and quantize it in each iteration while running the Q-learning algorithm; accordingly the controller must know the model.*

*(iv) Additionally, the initialization cannot be arbitrary (e.g. the initialization for the filter may be the invariant measure of the state under the exploration policy so that the iterations for the finite approximation given the initialization always remain in the absorbing set compatible with the invariant measure under exploration policy; this ensures that the infinite occupation conditions hold for the reachable quantized belief state and action pairs from the initialization).*

*(b) (i) For the finite window approach, a uniform convergence of filter stability, via $L_t$, is needed and it does not appear that only asymptotic filter stability can suffice.*

*(ii) On the other hand, this is a universal algorithm in that the controller does not need to know the model.*

*(iii) Furthermore, the initialization satisfaction holds under explicit conditions; notably if the hidden process is positive Harris recurrent, the ergodicity condition holds for every initialization; both the convergence of the algorithm as well as its implementation will always be well-defined.*

*For each setup, however, we have explicit and testable conditions. We note that it is possible to construct various discretization and approximation methods for POMDPs, and their corresponding belief counterparts. Each such approximation requires a different analysis to quantify the error of the approximation. See e.g. in Kara & Yüksel (2022) where a nearest neighbor approximation scheme is considered for finite memory information structures and the resulting loss function is in terms of more relaxed distance notions. Another alternative general direction is via so called 'approximate information states' (see Subramanian et al. (2022); Seyedsalehi et al. (2023)), where a uniform approximation error under various distance notions for the approximate information states for near optimality is considered.*

**Remark 3.6** (Further Models: Continuous-Time and Applications)**.** *We note that the richness of the convergence theorem manifests itself also in the applications involving continuous-time models Bayraktar & Kara (2023) where quantized Q-learning finds a natural application area, and applications to optimal zero-delay coding Cregg et al. (2023) which also studies several subtleties with regard to ergodicity of belief dynamics.*

### 3.5 Multi-Agent Models and Joint Learning Dynamics: Subjective Q-Learning and an Open Question

As our final application, we consider multi-agent models. Multi-agent reinforcement learning (often referred to as MARL) is the study of emergent behavior in reinforcement learning under multi-agent and complex environments, and is one of the important frontiers in artificial intelligence research. Consider an environment with $N$-agents, each of which generate actions, and whose rewards impact one another. Notably, for $i = 1, \cdots, N$,

$$x_{t+1}^i = f(x_t^i, u_t^i, u_t^{-i}, x_t^{-i}, w_t)$$

with cost criteria

$$\sum_t \beta^t c(x_t^i, u_t^i, u_t^{-i}, x_t^{-i})$$

or mean-field models with

$$x_{t+1}^i = f(x_t^i, u_t^i, \mu_t^N, w_t)$$

and sample path costs

$$\sum_t \beta^t c(x_t^i, u_t^i, \mu_t^N),$$

where

$$\mu_t^N = \sum_{k=1}^N \delta_{x_t^i}(\cdot)$$

We assume several information structures, for each $m = 1, \cdots, N$: (i) [Global state with local action] With $\mathbf{x} = \{x^1, \cdots, x^N\}$, we have $I_t^m = \{\mathbf{x}_{[0,t]}, u_{[0,t]}^m\}$, (ii) [Local state with local action and mean-field state] $I_t^m = \{x_{[0,t]}^m, \mu_{[0,t]}^N, u_{[0,t]}^m\}$, (iii) [Local state and local action] $I_t^m = \{x_t^m, u_{[0,t]}^m\}$. Accordingly, for each agent $u_t^m = \gamma_t^m(I_t^m)$ for all $t \in \mathbb{Z}_+$.

Given these policies, one would like to minimize the expected values of the cost functions defined above, and given the policies of other agents. Study of such decentralized systems is known to be challenging both for stochastic teams and stochastic games, where the cost functions above may depend on individual agents. Learning theory for such systems entails two primary challenges:

The first immediate challenge for learning in such models is due to decentralization of information: some relevant information will be unavailable to some of the players. The second difficulty inherent to MARL comes from the non-stationarity of the environment from the point of view of any individual agent. As an agent learns how to improve its performance, it will alter its behaviour, and this can have a destabilizing effect on the learning processes of the remaining agents, who may change their policies in response to outdated strategies. Notably, this issue arises when one tries to apply single-agent RL algorithms—which typically rely on state-action value estimates or gradient estimates that are made using historical data—in multi-agent settings. A number of studies have reported non-convergent play when single-agent algorithms using local information are employed, without modification, in multi-agent settings. Thus, for such models the main obstacle to convergence of Q-learning is due to the presence of multiple active learners leading to a non-stationary environment for all learners.

#### 3.5.1 Two-time scales and a Markov chain over play path graphs

To overcome this obstacle, also building on inspiration from prior work Foster & Young (2006); Germano & Lugosi (2007); Arslan & Yüksel (2017) modifies the Q-learning for stochastic games as follows: In the variation of Q-learning, DMs are allowed to use constant policies for extended periods of time called *exploration phases*. This is also referred to as *two-time scales* approach[1].

As illustrated in Figure 2, the $k-$th exploration phase runs through times $t = t_k, \ldots, t_{k+1} - 1$, where

$$t_{k+1} = t_k + T_k \qquad \text{(with } t_0 = 0\text{)}$$

---

[1]We note that an alternative two-time scales approach is via different learning rates applied by agents by taking advantage of slow learning (typically of policies) and fast learning (typically of values); see Borkar (1997; 2002); Leslie & Collins (2003; 2005); Sayin et al. (2021). This approach couples the stochastic learning dynamics with a system of ODEs which characterizes best-response dynamics, whose stability can be used to establish convergence in a variety of setups.

for some integer $T_k \in [1, \infty)$ denoting the length of the $k-$th exploration phase. During the $k-$th exploration phase, DMs use some constant policies $\pi_k^1, \ldots, \pi_k^N$ as their baseline policies with occasional experimentation.

The essence of the main idea is to create a stationary environment over each exploration phase so that DMs can almost accurately learn their optimal Q-factors corresponding to the constant policies (which is also slightly randomized to make room for exploration) used during each exploration phase and update their policies.

This machinery has been adopted under two types of policy updates: (i) *Best response dynamics with inertia* for weakly acyclic games Arslan & Yüksel (2017) considered for the case where each agent has access to the global state but only local state (requiring typically deterministic policies), and (ii) a variation of it which is referred to as *satisficing paths* dynamics Yongacoglu et al. (2022; 2023) which assumes that the agents have access to a variety of information states and the policies may be randomized.

Theorem 2.1, with the following perceived state updates for each agent, ensures convergence for each exploration phase, under the required conditions (see Yongacoglu et al. (2022)):

i [Global State] $S_t^m = \mathbf{X}_t, U_t = U_t^m, C_t = c(X_t^i, U_t^i, U_t^{-i}, X_t^{-i})$ (or $c(X_t^i, U_t^i, \mu_t^N)$ for the mean-field setup),

ii [Local State] $S_t^m = X_t^m, U_t = U_t^m, C_t = c(X_t^i, U_t^i, U_t^{-i}, X_t^{-i})$ (or $c(X_t^i, U_t^i, \mu_t^N)$ for the mean-field setup),

iii [Local and Mean-Field State or Compressed Global State] $S_t^m = \{X_t^m, F(\mathbf{X}_t)\}, U_t = U_t^m, C_t = c(X_t^i, U_t^i, U_t^{-i}, X_t^{-i})$, for some function $F$ (which may include $\mu_t^N = F(\mathbf{X}_t)$ as a special case).

**Subjective Satisficing Paths and Subjective Q-Learning Equilibrium**

Consider the following *subjective win-stay/lose-shift* algorithm: At the end of each exploration phase, if agents are $\epsilon$-satisfied, then they do not alter their policies. However, if they are not in an $\epsilon$-equilibrium, they randomly select a policy mapping their local perceived state to their actions, possibly with some inertia, where the policy space is quantized. In particular, the selected policies may be randomized (as they are not best responses or near best responses).

**Definition 3.2.** *Yongacoglu et al. (2022; 2023) Let $\epsilon \geq 0$ and let $\boldsymbol{\pi}^{-i} \in \boldsymbol{\Gamma}_S$. A policy $\pi^i \in \Gamma_S^i$ is called a $(\mathcal{V}^*, \mathcal{W}^*)$-subjective $\epsilon$-best-response to $\boldsymbol{\pi}^{-i}$ if*

$$V_{\pi^i, \boldsymbol{\pi}^{-i}}^{*i}(y) \leq \min_{a^i \in \mathbb{U}} W_{\pi^i, \boldsymbol{\pi}^{-i}}^{*i}(y, a^i) + \epsilon, \quad \forall y \in \mathbb{Y}.$$

**Definition 3.3.** *Yongacoglu et al. (2022; 2023) Let $\epsilon \geq 0$. A joint policy $\boldsymbol{\pi}^* \in \boldsymbol{\Gamma}_S$ is called a $(\mathcal{V}^*, \mathcal{W}^*)$-subjective $\epsilon$-equilibrium if, for every $i \in \mathbb{N}$, we have*

$$V_{\pi^{*i}, \boldsymbol{\pi}^{*-i}}^{*i}(y) \leq \min_{a^i \in \mathbb{U}} W_{\pi^{*i}, \boldsymbol{\pi}^{*-i}}^{*i}(y, a^i) + \epsilon, \quad \forall y \in \mathbb{Y}.$$

Yongacoglu et al. (2023) introduced such a paradigm and presented conditions under which equilibrium or subjective equilibrium is arrived at. The limit in which each agent is $\epsilon$-satisfied with respect to the computed value functions, as a result of the Q-learning iterations is referred to as a *subjective (Q-learning) equilibrium*.

Accordingly, each agent then applies (2) during exploration phases. This is stated explicitly in the following Yongacoglu et al. (2022):

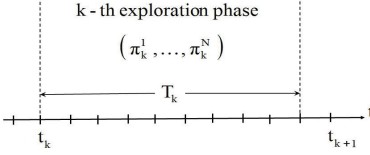

Figure 2: An illustration of the $k-$th exploration phase.

---

**Algorithm 1:** Independent Learning via $\epsilon$-Subjective Satisficing: Subjective Q-Learning Yongacoglu et al. (2022)

---

**1 Set Parameters**

**2**  $\Pi^i \subset \Gamma_S^i$ : a fine quantization of stationary policies $\Gamma_S^i : \mathbb{S} \to \mathcal{P}(\mathbb{U})$, where $s^i \in \mathbb{S}$, $u^i \in \mathbb{U}$

**3**  $\{T_k\}_{k \geq 0}$: a sequence in $\mathbb{N}$ of learning phase lengths

**4**   set $t_0 = 0$ and $t_{k+1} = t_k + T_k$ for all $k \geq 0$.

**5**  $e^i \in (0, 1)$: random policy updating probability

**6**  $d^i \in (0, \infty)$: tolerance level for sub-optimality

**7 Initialize** $\pi_0^i \in \Pi^i$ (arbitrary), $\widehat{Q}_0^i = 0 \in \mathbb{R}^{\mathbb{S} \times \mathbb{U}}$, $\widehat{J}_0^i = 0 \in \mathbb{R}^{\mathbb{U}}$

**8 for** $k \geq 0$ ($k^{th}$ exploration phase)

**9**  | **for** $t = t_k, t_k + 1, \ldots, t_{k+1} - 1$

**10**  | | Observe $s_t^i$

**11**  | | Select $u_t^i \sim \pi_k^i(\cdot | s_t^i)$

**12**  | | Observe $c_t^i := c(x_t^i, u_t^i, x_t^{-i}, u_t^{-i})$ and $s_{t+1}^i$

**13**  | | Set $n_t^i = \sum_{\tau = t_k}^{t} \mathbf{1}\{(s_\tau^i, u_\tau^i) = (s_t^i, u_t^i)\}$

**14**  | | Set $m_t^i = \sum_{\tau = t_k}^{t} \mathbf{1}\{s_\tau^i = s_t^i\}$

**15**  | | $\widehat{Q}_{t+1}^i(s_t^i, u_t^i) = \left(1 - \frac{1}{n_t^i}\right)\widehat{Q}_t^i(s_t^i, u_t^i) + \frac{1}{n_t^i}\left[c_t^i + \beta \min_{a^i} \widehat{Q}_t^i(s_{t+1}^i, a^i)\right]$

**16**  | | $\widehat{J}_{t+1}^i(s_t^i) = \left(1 - \frac{1}{m_t^i}\right)\widehat{J}_t^i(s_t^i) + \frac{1}{m_t^i}\left[c_t^i + \beta\widehat{J}_t^i(s_{t+1}^i)\right]$

**17**  | **if** $\widehat{J}_{t_{k+1}}^i(y) \leq \min_{a^i} \widehat{Q}_{t_{k+1}}^i(y, a^i) + \epsilon + d^i \; \forall y \in \mathbb{S}$, **then**

**18**  | | $\pi_{k+1}^i = \pi_k^i$

**19**  | **else**

**20**  | | $\pi_{k+1}^i \sim (1 - e^i)\delta_{\pi_k^i} + e^i \text{Unif}(\Pi^i)$

**21**  | **Reset** $\widehat{J}_{t_{k+1}}^i = 0 \in \mathbb{R}^{\mathbb{S}}$ and $\widehat{Q}_{t_{k+1}}^i = 0 \in \mathbb{R}^{\mathbb{S} \times \mathbb{U}}$

---

Theorem 2.1 shows that the exploration phase in Algorithm 1 is such that the two-time scale and satisficing-paths paradigm is applicable to a much broader class of setups.

Building on the general approach presented in Yongacoglu et al. (2022), it follows that under mild numerical parameter selection conditions, if (i) a subjective Q-learning $\epsilon$-equilibrium exists (with sufficiently fine quantization of the randomized stationary policy space) and (ii) if there is a finite $\epsilon$-subjective satisficing path from any initial policy profile to subjective Q-learning $\epsilon$-equilibrium equilibrium, Algorithm 1 will converge to a subjective equilibrium with arbitrarily high probability by adjusting the $T_k$ terms accordingly.

Beyond the setups in which the limit may be close enough to each agent's objective equilibrium (case with global state, or mean-field state information) Yongacoglu et al. (2022), and symmetric games Yongacoglu et al. (2023) conditions for the existence of subjective Q-learning equilibria is an open problem and requires further research. In particular, an application of Kakutani-Fan-Glicksberg theorem (Aliprantis & Border, 2006, Corollary 17.55) would entail a detailed study on the continuous dependence of the limit of Q-learning iterates in Theorem 2.1.

We hope that Theorem 2.1 will provide further motivation for research in this direction.

## 4  Conclusion

In this paper, motivated by reinforcement learning in complex environments, we presented a convergence theorem for Q-learning iterates, under a general, possibly non-Markovian, stochastic environment. Our conditions for convergence were an ergodicity and a positivity condition. We furthermore provided a precise characterization of the limit of the iterates. We then considered the implications and applications of this theorem to a variety of non-Markovian setups (i) fully observed MDPs with continuous spaces and their quantized approximations (leading to near optimality), (ii) POMDPs with a weak Feller continuity together

with a mild version of filter stability and quantization of filter realizations (which requires the knowledge of the model but more restrictive conditions on the initialization), (iii) POMDPs and the convergence to near-optimality under a uniform controlled filter stability plus finite window policies (which does not require the knowledge of the model and with an arbitrary initialization though under a more restrictive filter stability condition), and (iv) for multi-agent models where convergence of learning dynamics to a new class of equilibria, subjective Q-learning equilibria; where open questions on existence are noted. We highlighted that the satisfaction of ergodicity conditions required an analysis tailored to applications.

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
