# OpenReview forum: "Q-Learning for Stochastic Control under General Information Structures and Non-Markovian Environments"
_TMLR — Accepted by TMLR_

### Review · Reviewer_gfQB · 2023-12-21

**Summary Of Contributions:**

This paper presents a convergence theorem for stochastic iterations under a possibly non Markovian environment under ergodicity and positivity assumptions.
The work presents examples of applications.

**Audience:**

Yes

**Claims And Evidence:**

Yes

**Requested Changes:**

Requested Change:

1. Perhaps the introduction can be clarified more; some tables or figures showing the position of this work would help read the second question part in particular.  Also the author(s) mention Kara & Yuksel twice in the introduction, and it makes me feel the introduction flows less naturally; could be better to reorganize the paragraphs a bit (not necessarily a large edit though)

2. In the proof of Theorem 2.1. what is I_1?  Also in the theorem, how is the Q_0 assumed to be distributed?

3. Minor point; in page 7, with which is the learning rate linear?  In page 8 have you defined gamma?  3.5 and 3.6 subsections are related; should be within the same subsection?

4. Should put some texts describing how strong the ergodicity assumption is and show what kind of examples it does not apply.

(5. Can you elaborate more on the convergence rate in the future?)

**Strengths And Weaknesses:**

Strength:

The paper is clearly written and the message is clear as well.
Also, the unified view the work presents seems valuable.

Weakness:

It is not the weakness, but there seems a phrase that is revealing the author identity (it says “as we have studied in …” )

Should put some texts describing how strong the ergodicity assumption is and show what kind of examples it does not apply.

---

> ### Author Response · Authors · 2024-03-01
>
> We thank the referee for these comments. We have addressed the identity issue. We have elaborated on the ergodicity assumption in much further detail.
>
> We have updated the introduction which will hopefully help the reader to better understand the position and the contributions of the paper.
>
> $I_1$ is replaced by $s_1$ in the new version. The convergence holds independent of the initial $Q_0$. We have now added that the current version of the paper.
>
> The learning rate related part of the proof is revised; we have redefined the policy, and the Section 3 has been reorganized.
>
>
> Within the scope of the general result (Theorem 2.1), the ergodicity condition has two elements: that the process is uniquely ergodic and that the initialization is within the basin of attraction and finally the positivity condition. Some of these can be tested via general conditions; however some of them require the analysis tailored to an application. We have provided a detailed analysis in the paper especially in the context of POMDPs. The ergodicity assumption does not apply in its stated form for belief MDPs where the uniform quantization bins are not in the support of the invariant measure (for the measure valued belief process) under the exploration policy; however, one can tailor the analysis so that the approximation and convergence results hold.
>
> On the other hand, one can envision many scenarios where the process is not ergodic; such as those with infinite memory dependence. The ergodic theory for such infinite dimensional processes is an active area of research and we hope that our general result will find applications for such setups as well.
>
>
> * If a hidden environment under the exploration policy is Markovian, positive Harris recurrence (which reduces, for finite/countable models, irreducibility and positive recurrence) addresses these: the basin of attraction is the entire state space in this case. However, in the absence of such Harris recurrence one needs to cautiously study the initialization of the algorithm. If the process $S_t$ is Markov, then we just need to check the ergodicity of this Markov process $S_t$ under the exploration policy. As noted above, the positive Harris recurrence suffices (which reduces to for finite/countable models, irreducibility and positive recurrence); however if we only have unique ergodicity the initialization needs to be in the basin of attraction a restrictive version of which is that the initialization should be from a set of measure 1 under the invariant measure. If the process $S_t$ forms a Markov process together with some other process $X_t$, then we need to check the ergodicity of the joint process $(S_t,X_t)_t$ which is Markov; and the above applies.
>
> * When we we have a partially observed system where the observation and hidden state $(Y_t,X_t)_t$ form a Markov process and the belief/filter state itself is a probability measure valued Markov process. For this the ergodicity of the filter process is to be studied: Stability of $X_t$ together with filter stability implies unique ergodicity for the measure valued belief Markov process. The initialization is addressed once one starts the process according to the invariant measure via the arguments presented in the paper. On the other hand, for the finite window based approach for POMDPs, the ergodicity of $(Y_t,X_t)_t$ process is sufficient for the ergodicity analysis to hold (though at the expense of more restrictive geometric filter stability conditions).
>
>
> * A further special case is when we construct the Q learning with a local state of an agent $i$, say $x_t^i$ or local partial state $y^i_t$. In this case, the global state ${\bf x}_t=[x_t^i]_i$, will be a Markov process under Markov exploration policies or memoryless exploration policies, and thus, it is sufficient to check the ergodicity of the global state process.
>
> We thank the referee again for pointing out these.

---

> > ### Comment · Reviewer_gfQB · 2024-03-02
> > **Thank you for the response**
> >
> > Thank you for the response; the author(s) have addressed my concerns, and I'd recommend acceptance.

---

### Review · Reviewer_YkT4 · 2023-12-23

**Summary Of Contributions:**

This paper gives sufficient condition for Q learning to converge to a function satisfying a Bellman-type equation under a general set of ergodicity conditions that do not require the system to be Markovian. Then it is shown how these general conditions apply to a variety of non-Markovian settings, such as quantized MDPs, a few different approaches to POMDPs, and multi-agent problems.

**Audience:**

Yes

**Claims And Evidence:**

Yes

**Requested Changes:**

* Clean up the basic math issues described above.
* Articulate the contributions more explicitly and formally. (I.e. write theorems or corollaries for the applications of Theorem 2.1)
* Articulate more clearly what is gained from the current results beyond prior results, where appropriate.

**Strengths And Weaknesses:**

# Strengths

- The general theoretical result is a strong one, and compelling applications to interesting theoretical problems are given.

- The proof of the main result is clear, with some minor issues discussed below.

- The writing is well-organized, and the contributions are articulated well.

# Weaknesses

- Some of the formal statements are a bit awkward or sloppy:
    * In assumption 2.1, it is assumed that the ergodic average converges to the average with respect to some measure $\pi$, such that $\pi(B)>0$ for all measurable $B$. This does not make sense as the cost space is all of the real numbers, and so under any reasonable measure, there will be sets of measure zero. (More generally, the empty set should always have measure zero.). This assumption about $\pi(B)>0$ does not appear to actually be used, and as a result, it could likely be omitted or modified as needed.
    * A similar problem arises in Assumption 3.1
    * The limiting argument starting around equation (6) is a bit sloppily written. In particular, the iterates do not satisfy (6), and so it would be more precise to define a separate sequence with an equality, analogous to (6), and then show that the under the hypothesis that the iterates of interest converge to $M_1\ge C\epsilon$, then they must converge to the iterates of the separate sequence. Then show that the separate sequence must drop below $C\epsilon$.
- The contributions could be articulated more clearly in the main text.
    * All of the applications of Theorem 2.1 are stated as parts of paragraphs of text. This makes it a bit hard to see what the actual results are without careful reading. It would be better to formulate explicitly as theorems or corollaries, with clearly defined hypotheses.
    * Sometimes, it is difficult to disentangle the contributions of this particular work from prior work. This is particularly problematic in Section 3.3, in which all of the stated results are attributed to prior papers.

---

> ### Author Response · Authors · 2024-03-01
>
> We thank the referee for the encouragement and the detailed comments.
>
>
> We have revised the assumptions and fixed the issues about zero measured sets in Assumptions 2.1 and 3.1. We thank the referee for these corrections.
>
> We have updated the last part of the proof of Theorem 2.1 and made the analysis more explicit.
>
> We have updated the presentation of the technical results cautiously in view of the comments; the implications of our main theorem are now presented as Corollaries with several remarks to help guide the reader and aid with the flow. In the introduction we now explicitly emphasize the new results and the implications of the new results for the existing ones. These help us unify the results more effectively.

---

### Review · Reviewer_46oT · 2024-02-16

**Summary Of Contributions:**

This paper provides a convergence theorem for Q-learning under general non-Markovian stochastic environments based on ergodicity and a positivity criterion.  Then the implications of this theorem are explored in the context of stochastic control problems in non-Markovian environments: specific examples studied include quantized approximations of fully observed MDPs, belief-space MDPs (arising from POMDPs), and multi-agent models.  For multi-agent models in particular, convergence to subjective Q-learning equilibria is shown.  Furthermore, for POMDPs, under suitable technical assumptions, it is shown that finite memory policies can be used to arrive to near optimality.

**Audience:**

Yes

**Claims And Evidence:**

Yes

**Requested Changes:**

(1) Please make more explicit what is a novel result and what is an existing result that is derived in a new way;
(2) The paper is extremely dense, and at times, impenetrably terse.  Please consider adopting a less is more approach, and perhaps relegating some of the "existing results derived in a new way" to supplementary material, such that the novel results can be clearly explained and appreciated.
(3) Please consider running more experiments to empirically validate the theoretical results in the different environmental settings, or explicitly comment as to why this would not be useful or informative.

**Strengths And Weaknesses:**

S1) The technical contributions of the paper seem very solid and novel: presenting a unified analysis tool (Thm 2.1) for characterizing convergence criterion for Q-learning in the non-standard settings such as multi-agent models and quantized MDPs is very nice.
(S2) Theorem 2.1  strikes me as quite general and elegant, especially given the (relative) simplicity of its proof.
(S3) I appreciate the instantiation of Thm 2.1 on a simple example in section 2.1, as it helped understand its implications a bit better.
(S4) The analysis under non-traditional information structures in Section 3 is very interesting.  Providing convergence guarantees for POMDPs and multi-agent scenarios using a unified approach is very elegant.  However, the section would be made stronger if a clear delineation between what is novel and what is a recovery of existing results in the literature is made.  Indeed much of this section seems to be restating existing results from the literature.

(W1, minor) the writing style throughout is a bit odd, with very short paragraphs of just a single sentence or two.
(W2) I think the authors can do a better job of highlighting what is a truly novel result, and what is a novel derivation of existing results.
(W3) The notion of subjective equilibria is used prior to its definition (even informal) in the introduction.
(W4) While I appreciate the discussion at the start of Section 2 is meant to highlight the difference between the submitted paper and existing work, the discussion is very technical, and for a non-expert, it was difficult for me to extract what the significant differences were.  Would it be possible to have a table or something that summarizes which assumptions are made (or not made) in each of the respective papers (including the submitted one) so that the reader can identify at a glance where the proposed work fits in the existing work landscape?
(W5) The text is very dense, and there is very little prose to help the reader interpret the consequences of, e.g., Assumptions 2.1 and 2.2.
(W6) Controlled Markov chains are only defined in equation (8), at the start of section 3, and yet are mentioned in section 2.1.  I would suggest a reorganization of preliminary material so that concepts are introduced/defined before they are referred to in the text.

---

> ### Author Response · Authors · 2024-03-01
>
> We thank the referee for the detailed comments and positive assessment.
>
>
> We have updated the paper cautiously in view of the comments; the implications of our main theorem are now presented as Corollaries with several remarks to help guide the reader and aid with the flow. In the introduction we now explicitly emphasize the new results and the implications of the new results for the existing ones. We discuss the conditions in further detail and make comparisons more explicitly (see e.g. Remark 1 on the finite MDP setup, Remark 3.3 when compared with prior work imposing positive Harris recurrence, and Remark 3.5 on the POMDPs under the two methods presented). We have revised the introduction section based on the comments to highlight the technical relaxations.
>
> We have tried to make the paper more accessible to read and have aligned the main result to be the unifying theme in the paper.
>
> We intend to leave extensive simulation based analysis for further studies and we hope that the current simulation serves as a useful case study. Indeed, there are also many real applications, such as applications in health/medical sciences e.g. optimal insulin therapy problem for a patient viewed as a POMDP; it would be desirable to study such important setups in the future. We hope that our paper will provide inspiration for mathematical approaches for such applications.

---

### Decision · Action_Editor_hFou · 2024-04-16

**Recommendation:** Accept as is

**Comment:**

This is a very strong theory paper, and it deserves recognition on that basis.

**Audience:**

Yes, this work is perhaps most relevant to RL theorists, but the results are applied in diverse enough situations to yield interest to many other groups (e.g. deep RL practitioners, multi-agent researchers, etc).

**Claims And Evidence:**

All of the reviewers agreed, even before the rebuttal period, that this paper's technical contributions were well supposed and would be of interest to the TMLR community. There was some minor points around clarifying aspects of a few of the proofs and writing flow, but even these small issues were addressed satisfactorily.